# An Up-to-Date Narrative Review on Congenital Heart Disease Percutaneous Treatment in Children Using Contemporary Devices

**DOI:** 10.3390/diagnostics12051189

**Published:** 2022-05-10

**Authors:** Stefana Maria Moisa, Alexandru Burlacu, Crischentian Brinza, Elena Țarcă, Lăcrămioara Ionela Butnariu, Laura Mihaela Trandafir

**Affiliations:** 1Department of Pediatrics, Faculty of Medicine, “Grigore T. Popa” University of Medicine and Pharmacy, 700115 Iasi, Romania; stephaniemed@yahoo.com (S.M.M.); laura.trandafir@umfiasi.ro (L.M.T.); 2Faculty of Medicine, “Grigore T. Popa” University of Medicine and Pharmacy, 700115 Iasi, Romania; crischentian.brinza@d.umfiasi.ro; 3Institute of Cardiovascular Diseases “Prof. Dr. George I.M. Georgescu”, 700503 Iasi, Romania; 4Department of Surgery II—Pediatric Surgery, “Grigore T. Popa” University of Medicine and Pharmacy, 700115 Iasi, Romania; 5Department of Mother and Child Medicine–Genetics, “Grigore T. Popa” University of Medicine and Pharmacy, 700115 Iasi, Romania

**Keywords:** pediatric cardiology devices, interventional procedures, novel devices

## Abstract

Background: Congenital heart pathology has a significant burden regarding morbidity and mortality in the pediatric population. Several transcatheter interventions and devices have been designed as an alternative to surgical repair. Percutaneous interventions have been proven to yield good results in most cases but with less stress and trauma than that attributed to surgical treatment, especially in frail pediatric patients. We aimed to review the literature and to investigate the feasibility and efficacy of transcatheter interventions and implantable devices for congenital heart disease management in children. Methods: We performed a search in Scopus and MEDLINE databases using prespecified keywords to retrieve clinical studies published between 2000 and 2021. Results: This article provides an up-to-date review regarding the applicability of interventional techniques in simple inter-atrial or inter-ventricular defects, and in challenging congenital defects, such as hypoplastic left heart syndrome, tetralogy of Fallot, or coronary artery fistula. Furthermore, we reviewed recent indications for defibrillator and cardiac resynchronization therapy, and new and promising devices currently being tested. Conclusion: Transcatheter treatment represents a feasible and efficient alternative to surgical repair of congenital heart defects. Novel devices could extend the indications and possibilities of percutaneous interventions in pediatric patients with congenital heart diseases.

## 1. Introduction

About 0.8–1.2% of all children born each year worldwide suffer from congenital heart defects, with isolated ventricular septal defect being the most frequent type. Moderate and severe forms of disease account for up to 6 in 1000 cases [1]. The survival of these children was dramatically improved during the last decades with the use of medical, interventional, and surgical methods. Devices are often used off-label because extensive studies have not yet been performed in children. Medication is also used in this manner, with up to 67% of the drugs used in the intensive care having limited or no approval for pediatric use [2] and up to 78% of congenital or acquired heart disease children receiving 1 or more off-label prescriptions in the late 2000s [3].

Percutaneous device treatment opened an entirely new era for congenital heart defect children, including those with isolated heart defects (pulmonary valve stenosis, patent ductus arteriosus (PDA), aortic coarctation, isolated inter-ventricular and inter-atrial communications), and those with complex heart disease (hypoplastic left heart syndrome (HLHS) and tetralogy of Fallot (ToF)). Percutaneous device treatment refers to transcatheter management of congenital heart defects using approved devices. One study investigated device failure associated with ventricular septal defect closure and found that 7.6% of cases treated in this manner encountered device failure [4]. No association was found between poor evolution and patient age, ventricular septal defect type, or device size or type. Others documented successful percutaneous closure of aortopulmonary windows [5], pulmonary arteriovenous fistulas [6], or ruptured Valsalva sinus aneurysms [7].

Amplatzer devices have been successfully used for atrial septal defect closure since 1997, proving to be safer than surgical interventions [8,9]. PDA interventional closure was also proven to be effective, with no complications in most cases [10]. A previous study documented 5 cases of postprocedural complications out of 407 children who underwent this procedure and concluded that large atrial defect size is a good predictor of subsequent complications [11].

The safety and benefit of balloon atrial septostomy has been proven in the transposition of great arteries and could also prove beneficial in both types of atrioventricular valve atresia and in total anomalous pulmonary venous return with no or restrictive foramen ovale [12]. Interventional procedures have also been successfully used in the multistep palliation of HLHS [13] or even for the full correction of certain variants of ToF [14].

Some important barriers in pediatric transcatheter treatment innovations have been described [15]. One of the most common barriers constitutes a reluctance regarding the testing and validation of novel devices in this pediatric population due to a greater concern regarding the legal or ethical aspects of the research. Moreover, conflicting knowledge related to recommendations for the development and manufacturing of pediatric devices could delay the evolution of percutaneous treatment [15]. Notwithstanding all the difficulties and barriers, novel pediatric devices have emerged as a consequence of technological progress [16].

We aimed to review the literature and to investigate the feasibility and efficacy of transcatheter interventions and contemporary implantable devices for congenital heart disease management in children. Consequently, we provide an updated narrative review integrating various percutaneous treatment techniques and devices used in pediatric interventional cardiology.

## 2. Interventional Cardiology Devices

Several types of devices are routinely used by pediatric cardiac interventionists while others are seldom used, as they do not have such a wide applicability in children (Table 1) [17]. Devices are used due to being approved by the competent organisms and in an off-label manner. In one study that enrolled 473 patients, the most used devices for transcatheter treatment were dilation balloons, occlusion devices, embolization coils, and stents. Other used percutaneous devices were relatively uncommon [18].

These devices are used for their approved indications but also in an off-label manner. For instance, radiofrequency perforation wires are approved for atrial septostomy but are also used for atretic valve perforation. Embolization coils are used to occlude small patent ductus arteriosus off-label (being approved for arterial, venous, and arteriovenous fistulae embolization) while dilation balloons, approved for valvar pulmonary stenosis treatment, are also used for stenotic aortic valves, stenotic conduits, or aortic coarctation. A cutting balloon can be used not only for desobstruction of a dialysis fistula but also for atrial septostomy or pulmonary artery stenosis treatment while stents can be placed off-label in a stenotic conduit, a stenotic pulmonary or even systemic vein, a coarctation site, a stenotic pulmonary artery branch, or can be used to keep patent ductus arteriosus in an open state [17]. These off-label prescriptions are not without their risks, such as residual shunts, coil embolism, or intravascular hemolysis [19]. Transcatheter closure of patent ductus arteriosus is possible using Gianturco coils in adolescents and adults [20]. Occlusion devices are approved for patent ductus arteriosus, muscular ventricular septal defect, ostium secundum atrial septal defect, and Fontan fenestration closure [14]. Any attempt to solve a Blalock Taussig shunt occlusion or a fenestration occlusion or a Gerbode shunt in an interventional manner is considered off-label [21].

## 3. Specific Interventions in Pediatric Cardiology

### 3.1. Pulmonary Valve Atresia or Critical Stenosis: Interventions and Devices

Although pulmonary valve atresia with intact ventricular septum is a relatively rare congenital heart disease, these patients are at an increased mortality risk, even after surgical intervention (52% in the first year) [22]. Different percutaneous techniques were developed to improve clinical outcomes, including balloon valvulotomy, ductus arteriosus stenting, or systemic-to-pulmonary shunt creation. The last two interventions are palliative ones, ensuring only univentricular circulation [23].

Most studies in the literature reported similar techniques of pulmonary valve perforation and balloon valvulotomy. A 4F (or 5F) right coronary catheter was advanced towards the infundibular area in one study. Afterward, different guidewires were used to perforate an atretic pulmonary valve: the stiff end of a 0.014-inch guidewire, 0.018-inch, or 0.024-inch radiofrequency guidewires [24]. Once the perforation was accomplished, a 0.014-inch guidewire was placed in the femoral artery and snared through a 4F femoral sheath. The pulmonary valve was then dilated using balloons with progressively increasing diameters. Some authors used balloons with a 20% larger diameter than the pulmonary annulus [24] while others used up to 1.2–1.4 larger balloon diameters [23]. In patients referred for ductal stenting as a palliative measure, femoral access is usually required [23].

Early data supported percutaneous pulmonary valvulotomy in patients with a well-formed right ventricular infundibulum, pulmonary valve annulus ≥ 7 mm, and tricuspid valve annulus ≥ 11 mm [25]. In addition, some authors excluded patients with documented coronary artery to the right ventricle fistula, but others did not consider it a procedural contraindication [23,24].

### 3.2. Patent Ductus Arteriosus—Percutaneous Closure

Pediatric patients with PDA and favorable anatomy for interventional treatment can be referred for percutaneous closure in the first year of life [26,27]. In the last years, various occlusion devices have become available, which has extended the possibilities of PDA percutaneous closure: Amplatzer Duct Occluder I, Amplatzer Duct Occluder II, Amplatzer Piccolo Occluder, Amplatzer Vascular Plug II, Nit-Occlud, Cera PDA Occluder, and Gianturco coil (Figure 1) [28,29].

The early available Amplatzer-type devices were not indicated in patients with low body weight. However, the Amplatzer Piccolo Occluder could be used even in patients ≥ 700 g and was approved by the Food and Drug Administration in 2019 [29]. Due to a higher risk of protrusion into the aorta and pulmonary artery, significant PDA was considered a contraindication for percutaneous closure using first-generation Amplatzer Occluder devices. Nevertheless, in 1 study, large PDAs (narrowest diameter of 4.1 ± 1.1 mm) were occluded successfully using Amplatzer Duct Occluder II devices in patients < 3 years old [30].

Usually, when an Amplatzer device is considered for PDA closure, ductus arteriosus is crossed from the venous side towards the aorta. In most studies, the device size was 1–2 mm greater than the narrowest diameter of ductus arteriosus. The device is positioned in the descending aorta through a long sheath. Initially, the distal disc should be opened and placed on the ductal aortic side. After confirming the proper position of the system, the occluder device could be delivered [31]. Nit-Occlud is a nitinol coil device approved for percutaneous closure of PDAs < 4 mm in diameter for patients > 5 kg aged ≥ 6 months. The device requires a 4F or 5F catheter and is also inserted from the pulmonary side of the PDA [32].

### 3.3. Aortic Coarctation—Percutaneous Intervention

Percutaneous balloon angioplasty with or without stent implantation represents an alternative to surgical intervention, even in infants [33]. Balloon angioplasty is usually performed in younger patients with low body weight (≤20 kg), as a larger sheath is needed for stent deployment. In one study, low-weight patients who underwent aortic coarctation stenting had good long-term outcomes but increased risk of femoral artery occlusion [34]. On the other hand, balloon angioplasty proved to be efficient, especially in the acute setting, with a low rate of adverse events in patients aged 3–12 months [33]. Stent implantation appeared to be safe in patients aged >1 year, irrespective of body weight (<30 kg vs. ≥30 kg) [35]. Notably, there are currently available stents that could be further dilated as patients grow (IntraStent LD Max TM could be dilated to 24–26 mm compared to Genesis stents, which could be dilated to 18–20 mm) [36].

In selected patients for percutaneous treatment, venous and arterial femoral access should be performed [37]. The proximal and distal diameters of the aorta and stenosis length and the length between the coarctation and aortic arch branches must be measured prior to angioplasty. The balloon and stent should be sized according to the proximal aortic diameter as it could co-exist as a post-stenotic dilation [36]. To enhance the balloon and stent passage through the stenotic area, a long stiff wire (Amplatz Super-Stiff, Rosen) can be placed distally in the subclavian artery or snared through a brachial approach [36].

Typically, a 1–2 mm larger balloon than the diameter of the pre-stenotic aorta is used for percutaneous angioplasty. Before stent deployment, some authors recommend testing stenosis distensibility by inflating the balloon at low pressure (<4 atm). The stent might not fully expand in highly fibrotic stenosis after the first intervention. Once patients grow, additional balloon inflations could be performed to expand the stent further [36].

Although the immediate success rate is high (97% even in low-weight children), some complications should be addressed [34]. In a previous study that enrolled children ≤ 20 kg, the rate of interventional procedure-related complications was 18%, with femoral artery injury being the most frequent [34]. Other potential complications include stent fracture, brachiocephalic vessel occlusion, aortic wall injury (tear, dissection, aneurysm), stroke, and peripheric embolization. Aortic dissection might require the deployment of a second stent, including a covered stent, or surgical intervention [36].

### 3.4. Coronary Artery Fistula—Percutaneous Management

Congenital coronary artery fistula (CAF) is a rare condition in the general population, with a 0.002% reported incidence [37]. Besides the congenital etiology, CAF can develop following surgical and interventional procedures (heart transplant, septal myomectomy, percutaneous coronary intervention, infection) [38]. Although spontaneous closure was reported in the literature (28.9% after 21 months), more than half of CAF becomes larger in time [38,39]. In addition, CAF can cause several symptoms and complications due to left-to-right or left-to-left shunts, including heart failure symptoms, ischemia, endocarditis, and cardiac arrhythmias [37]. Therefore, surgical or percutaneous closure in selected patients might be required [40].

Even though the right coronary artery was the most common origin site, CAFs can emerge from the left main, left anterior descending, or circumflex coronary arteries, as documented in one study [37]. CAFs were observed to drain in a cardiac chamber (right or left atrium, right or left ventricle) and in superior vena cava or a main pulmonary artery [37,38]. Percutaneous closure feasibility depends on the anatomical and technical features. An interventional approach can be considered in patients with a single CAF diameter greater than 2 mm, a straight pathway, and narrow drainage [37,38].

Different techniques of CAF occlusion have been described, highlighting the anatomical variations (origin, pathway, drainage): antegrade venous approach (long CAF), retrograde arterial approach (small and medium-sized CAF), arterio-arterial circuit approach (CAF draining in the left atrium or ventricle), and arterio-venous circuit approach (CAF draining in the right atrium or ventricle) [37]. According to the CAF size, length, and drainage, several devices can be used for percutaneous occlusion, as was performed in one study: duct occluder, muscular ventricular septal occluder, vascular plugs, and coils (Figure 1) [37]. In larger CAFs, some authors reported successful use of the Amplatzer Duct Occluder in percutaneous occlusion, with good short- and long-term results [41]. Small residual shunts might spontaneously close after several months from the index procedure [38].

### 3.5. Ventricular Septal Defect—Interventions and Devices

Isolated ventricular septal defect (VSD) constitutes the most frequent form of congenital heart disease, with an incidence of 2–5% in newborns. However, although the incidence is high, most small VS (almost 90%) close spontaneously within 1 year. Percutaneous correction is reserved for pediatric patients with hemodynamically significant VSD, including left heart volume overload or pulmonary to systemic flow ratio ≥ 2 [40]. Nevertheless, a previous study included pulmonary to systemic flow ratio ≥ 1.5 as an eligibility criterion [42]. A scientific statement from American Heart Association (AHA) does not recommend percutaneous closure of VSD in pediatric patients with inadequate anatomy for device implantation and asymptomatic cases without pulmonary hypertension [40].

Device closure of VSD in selected pediatric patients was associated with a reduction in left ventricular dimensions, volumes, and mass at 6 months (*p* < 0.05 for all), with a reduced risk of significant complication [43]. Some authors defined a too-large VSD for percutaneous closure as a defect requiring a device and a sheath larger than those recommended for the bodyweight [42]. The device should be sized 1–2 mm larger than the VSD diameters. Moreover, percutaneous closure techniques and devices were designed for muscular and perimembranous VSD, with a sufficient length between the defect and cardiac valves [42].

Through the femoral arterial access site, a catheter (e.g., right coronary or Amplatzer right coronary catheter) is advanced through VSD, and a guidewire is snared using the venous access site (femoral or jugular), thus establishing a loop [4,42]. Afterward, a device of appropriate dimensions is advanced and deployed after confirming the proper position. Various devices have been used in clinical studies involving the pediatric population: Amplatzer muscular and perimembranous VSD occluder, Amplatzer Duct Occluder I and II, LifeTech ductal device, and Nit-Occlud Lê VSD coil (Figure 1) [4,42,44].

### 3.6. Atrial Septal Defects—Percutaneous Device Closure

Atrial septal defect (ASD) constitutes another common congenital heart disease, affecting 1 patient from 1000 live births [45]. Percutaneous ASD closure has replaced surgery in many dedicated centers due to the availability and approval of occluding devices in secundum ASD [40]. Thus, percutaneous intervention for hemodynamically significant ASD in pediatric patients received class I indication in AHA guidelines [40]. Moreover, transcatheter ASD treatment can be considered in symptomatic patients with a transitory right to left shunt. However, percutaneous closure is not recommended in other types of ASD than ostium secundum and patients with advanced pulmonary vascular disease [40].

Transcatheter treatment of ASD was highly efficient even in low-weight children (<15 kg), as observed in one study [46]. The most used device for ASD occlusion was the Amplatzer septal occluder, followed by the Gore HELEX septal occluder, CardioSeal septal occluder, and Amplatzer multifenestrated cribriform septal occluder (Figure 2). Even though shunt resolution was obtained in 99% of patients during follow-up, the rate of major short-term complications was 5.5%. Therefore, transcatheter treatment should be delayed in asymptomatic patients with small defects until 4–5 years [46]. Other devices investigated in clinical studies included MemoPart ASD (self-centering) [47], Nit-Occlud ASD-R [48], Occlutech Figulla [49], CeraFlex [50], Cardi-O-Fix occluders [51], and Amplatzer Trevisio occluders [52].

Transcatheter ASD closure using double-disk devices is similar across clinical studies. After obtaining a femoral venous access site, a multipurpose catheter is advanced through ASD, and a guidewire is placed in the left upper pulmonary artery [53,54]. A sizing balloon can be used to measure the defect diameter, complementary to that obtained by transesophageal echocardiography. The recommended sizing technique is based on balloon inflation until the trans-septal flow stops, as observed echocardiographically, to avoid overestimation of the defect diameter. The balloon diameter is then measured, and an appropriately sized device is chosen and deployed [53,55].

### 3.7. Gerbode Ventriculo-Atrial Defect—Insights from Transcatheter Closure

A Gerbode defect characterizes the communication between the left ventricle and right atrium. A Gerbode defect can be congenital (a rare form of VSD) or acquired (cardiac surgery, aortic or mitral valve implantation, endocarditis) [56,57]. Surgical intervention was the standard of care for Gerbode defects. Transcatheter treatment emerged as an alternative to cardiac surgery once different percutaneous occluding devices became available, even though data are limited to case series and case reports [57].

Some of the most frequent devices used to occlude Gerbode defects in children are represented by the Amplatzer Duct Occluder II and Nit-Occlud Lê VSD coil [56,58,59]. Irrespective of the catheters and guidewires used, arterial femoral access is required. A catheter is then retrogradely advanced through the defect from the left ventricle to the right atrium. Afterward, a guidewire is snared and externalized using a venous femoral approach, thus establishing an arterio-venous loop. The occluding device is then advanced and deployed after confirming the proper position [56].

One study, which enrolled eight children with congenital and acquired Gerbode defects, confirmed the efficacy of percutaneous closure using a Nit-Occlud Lê VSD coil. Although a small residual shunt was observed in six patients, it was entirely closed in almost all patients during follow-up. However, a minor persistent defect was observed in one patient. Moreover, no patient developed cardiac conduction abnormalities, an important complication after surgical treatment [56].

### 3.8. The Blalock–Taussig Shunt—Rationale for Percutaneous Closure

The Blalock–Taussig systemic to pulmonary shunt represents a palliative surgical intervention in various congenital heart diseases. It can also be performed as a bridge therapy until surgical or interventional correction is performed [60,61]. However, patients can develop, in evolution, symptoms of heart failure due to pulmonary fluid overload. Usually, surgery is required to close the Blalock–Taussig shunt when indicated. However, when a surgical clamp cannot be used in selected patients, percutaneous occlusion represents a valid option of treatment [61].

In one study, an Amplatzer Vascular Plug device was used for transcatheter closure of the Blalock–Taussig shunt [61]. The end of the shunt was cannulated, and a suitable coronary guidewire was placed distally inside the shunt. Afterward, the device was deployed with good results during follow-up in a series of three patients [61]. Gianturco coils were also used successfully to occlude the Blalock–Taussig shunt in another study. Coils were sized 1–2 mm greater than the shunt diameter. If the shunt flow persisted after the first coil, supplementary coils were deployed. The shunt remained occluded during long-term follow-up (median of 3 years) [62].

Moreover, a shrinking technique with stents implanted in the graft was described in symptomatic patients with a too-large shunt. In one case report, the authors implanted 4 stents (Genesis XD), thus achieving a narrower shunt lumen and improved symptoms at 4 months of evaluation (Figure 1) [63].

### 3.9. Tetralogy of Fallot—Between Palliative and Full Percutaneous Correction

ToF is a common and complex congenital heart disease, with surgical treatment being the standard of care [14]. Percutaneous intervention might be required before open-heart surgery as a palliative measure. Nevertheless, once different techniques and devices were developed, full transcatheter correction became an attainable objective [64]. In addition, percutaneous interventions can be performed after surgical correction to treat residual shunts or long-term complications (e.g., pulmonary valve stenosis) [64].

In patients with ductus-dependent pulmonary flow, ductus arteriosus percutaneous stenting can be performed as an initial palliative intervention [65]. Different transcatheter techniques can be used according to the ductal origin, tortuosity, and length. A femoral venous approach might be required in cases with PDA emerging from the transverse aortic arch while an arterial approach can be performed in PDA emerging from the descending aorta [66]. A long sheath is placed near the ductal origin in the retrograde approach. PDA is crossed using a 0.014-inch guidewire through a right coronary catheter. The PDA length should be accurately measured to ensure appropriate stent sizing. Some authors used patients’ weight to guide stent diameter selection [67]. Although coronary stents were implanted in some clinical trials to maintain PDA flow, currently, more flexible self-expanding stents are available that could be used in this context [66].

When a femoral venous approach is chosen, a catheter is advanced from the right heart towards the aortic arch across the VSD. Different catheters can be exchanged to obtain a good position near the ductal ampulla required for stent passage [67]. Rarely, ductus arteriosus might emerge from the left or right subclavian artery, with a similar vertical trajectory to PDA originating from the aortic arch. In this case, the antegrade arterial approach is preferred. Usually, a two-stent technique is preferred to cover both the distal and the proximal segments of the PDA [67].

Other interventions for ToF involve right ventricular outflow tract (RVOT) obstruction management. In one study, the authors reported successful implantation of two Melody valves in a pediatric patient with TOF and acquired double RVOT with good clinical outcomes during long-term follow-up (7 years) [68]. Percutaneous interventions for pulmonary valve stenosis and VSD were described above.

### 3.10. Hypoplastic Left Heart Syndrome—Transcatheter Interventions

Percutaneous treatment for HLHS constitutes a palliative intervention before surgical correction. The transcatheter intervention aims to delay the surgical intervention, as open-heart surgery might have worse outcomes [69].

As a first step, the palliative intervention includes pulmonary artery flow restriction and ductus arteriosus stenting to obtain satisfactory pulmonary and systemic blood flows. A hybrid approach and a full percutaneous technique were described in this setting [69,70].

In one study investigating the hybrid approach, pulmonary artery banding was achieved through median sternotomy. Afterward, a 0.018-inch guidewire was used to cross the ductus arteriosus and was placed in the distal descending aorta. A sinus-SuperFlex stent, 1–2 mm larger than the ductus arteriosus measured diameter, was implanted. The authors reported an 83% survival rate during 137 days [71].

A relatively new entirely percutaneous technique was described and investigated in clinical studies enrolling HLHS patients. In this regard, a venous femoral access site is obtained, as reported in one study. Modified Micro Vascular Plug (MVP) devices were used as flow restrictors in the pulmonary arteries. The modified MVP replaced surgical pulmonary banding, avoiding inherent surgical risks. The ductus arteriosus was then crossed using a guidewire, and a Formula stent was deployed [69]. A similar transcatheter procedure was performed in six neonates in another study. Ductal stenting and endovascular modified MVP implantation were also used to restrict pulmonary flow and achieve an adequate systemic flow (Figure 3) [71].

### 3.11. Balloon Atrial Septostomy—Rationale and Technique

Balloon atrial septostomy constitutes a palliative intervention before lung transplantation in patients with pulmonary arterial hypertension and patients with HLHS, tricuspid, or pulmonary valves atresia. However, balloon atrial septostomy is efficient if performed in neonates or children aged < 6 weeks [72,73,74]. Other techniques were developed for older children, such as blade atrial septostomy, radiofrequency perforation, or trans-septal puncture [40].

A venous access site (femoral or umbilical vein) should be obtained. Several available catheters can be used to cross foramen ovale: the Miller catheter, Rashkind balloon catheter, Fogarty balloon catheter, and NuMED septostomy catheter (Figure 3). The balloon is inflated in the left atrium and pulled back into the right atrium. After balloon deflation, the procedure can be repeated several times (four to six times as reported in the literature). In the case of the Miller septostomy catheter, the balloon is traditionally inflated with 4–6 mL of fluid. Fluoroscopic balloon position should be confirmed before inflation to avoid complications [40,75]. In one study, balloon septostomy was linked to good early outcomes, atrial communication diameter and oxygen saturation increased, and the mean pressure gradient significantly decreased [75].

Even though balloon atrial septostomy is highly efficient, some complications can occur: balloon rupture or embolization, mispositioned balloon, cardiac perforation, vascular access site injury, or stroke [40].

### 3.12. Inferior Vena Cava Filters in Children—How and Why

In the last two decades, deep venous thrombosis (DVT) and embolism have been reported more frequently in children. DVT might occur secondary to oncologic therapy, intensive care unit stay, or trauma in children. In this context, inferior vena cava (IVC) filter placement could replace anticoagulant therapy in selected patients (including children in whom anticoagulation is contraindicated) and could be performed as an adjunctive therapy. Some authors used IVC filters in patients considered for local thrombolysis for DVT or prior to high-risk surgical interventions [76,77].

One study investigated the long-term efficacy and complications related to Greenfield IVC filters in 15 children. During 16 years of follow-up, none of the patients developed pulmonary embolism. Mild femoral venous reflux was observed in three patients but without venous obstruction [78]. Notably, retrievable IVC filters (e.g., Günther Tulip filter) might also be considered in pediatric patients [76,79]. The authors repositioned the IVC filter several times before removal in one study, with promising efficacy and safety profiles [79].

Before IVC filter implantation, a venous access site (femoral, jugular) should be obtained. Using a catheter positioned in the infrarenal IVC, a guidewire is advanced and placed distally. After confirming the position, the IVC filter is deployed [79]. Potential acute and late complications linked to IVC filter placement include IVC perforation, endovascular infection, filter migration, stenosis, or thrombosis. Nevertheless, in high-risk selected patients, the benefits of IVC filter placement outweighed the risk of procedural complications [76].

### 3.13. Pacemakers and Defibrillators

The first pediatric pacemaker implant was performed in November 1960 by Dr. Zoll and Dr. Frank in a child who had developed complete heart block following a ventricular septal defect repair procedure [80]. Nowadays, this procedure is routine in some centers and pacemakers are implanted for sinus bradycardia, the tachycardia-bradycardia syndrome, and second or complete atrioventricular block [81].

Some of these centers also perform biventricular pacing and implantable cardioverter-defibrillator (ICD) implants. ICDs are implanted in children for the primary or secondary prevention of life-threatening arrythmia [82]. Children with long QT syndrome, sustained ventricular tachycardia, resuscitated cardiac arrest, catecholaminergic polymorphic ventricular tachycardia, glycogen storage disease-related cardiomyopathy, hypertrophic cardiomyopathy, uremic cardiomyopathy, and various congenital heart diseases with syncope can benefit from ICD implantation [83]. Those who undergo full repair of Fallot tetralogy [84] or transposition of great arteries or congenital aortic stenosis are at higher risk for sudden cardiac death [85].

These children, with complicated intracardiac anatomy, are better suited for epicardial pacing. In a study performed by Rico-Mesa et al., the most frequent complications associated with intracardiac pacing or defibrillation were hematoma, infection, and pneumothorax, followed by venous occlusion, embolism, or death [86]. The same study described device-related complications (for example, electrode fracture) more frequently with biventricular devices (26.7%) compared to ICD (11.5%) and pacemakers (7.2%). Patient-related complications were also more frequent with biventricular devices (19.4%) than ICDs (5.9%) and pacemakers (11.2%).

Biventricular assist devices do not have clear implantation indications and guidelines in children; however, they seem to improve systolic function in children with cardiomyopathies or congenital heart diseases by 12.8 ± 12.7%, thus delaying heart transplantation indication [87]. Children with dilated cardiomyopathies should receive cardiac resynchronization-defibrillator (CRT-D) devices while those with atrioventricular septal defects and complete atrioventricular block could benefit from cardiac resynchronization with biventricular pacemaker (CRT-P) therapy [88].

## 4. Novel Devices

Novel devices that can be applied to pediatric cardiology patients are being developed every day. In 2018, The Pediatric Device Innovation Symposium revealed several gems [16]:The CAM (Carnation Ambulatory Monitor) is an ECG monitor that can store 7 days’ worth of continuous stripe. The monitor is small, and it attaches to the patient’s sternum to capture high-resolution ECGs that can be uploaded to a web page afterwards. This device is already available for clinical use [89].Children undergoing open heart surgery could soon benefit from the VentriFlo True Pulse pump, an extracorporeal blood pump that is able to provide a complete stroke volume while using a flexible membrane and valves. This device has already obtained its patent [90].Another novel and promising device is the LEAP valve, a prosthetic heart valve that “grows with the child” by expanding up to twice its initial size. This device is still undergoing ex vivo tests and has not yet reached the animal testing phase [91].Its main competitor seems to be PolyVascular, another expandable polymer valve, which is also not yet available for use [92].Another great help in terms of mechanical circulatory support seems to be the NuPulseCV intravascular ventricular assist system, which is inserted in the subclavian artery. The main benefits of this system are that it is minimally invasive and can be used in a long-term manner. This device is already in its human testing phase, which begun in adults in 2016 and is currently still recruiting [93].ExGraft is an expendable radiopaque synthetic vascular graft that accommodates the child’s growth and reduces the need for repeated surgery. This device is currently available in the European Union [94].The FloWatch pulmonary artery banding device has long surpassed its testing phase. Its previous version was used between 2002 and 2012. The new version of this device is telemetry controlled to allow calculated amounts of blood to flow through the pulmonary artery [95].Children suffering from end-stage kidney disease could benefit from The Arteriovenous Fistula Eligibility (AFE) system use. This is an external blood pump that stimulates venous dilation and improves the likelihood for successful placing of an arteriovenous fistula, and has not yet undergone human studies [96].Arterial lines and inflatable cuffs could soon be replaced by the Boppli blood pressure monitor, which monitors blood pressure in a noninvasive, continuous, and comfortable manner. The device looks like a watch and is not yet available for use [97].

## 5. Perspectives

This paper opens the pathway to future research related to pediatric cardiology. Future work should not only include a thorough follow-up on the novel devices discussed here but also on new interventional technique development and detailed research on nanotechnologies applicable to newborn, infant, and pediatric cardiac malformations.

## 6. Conclusions

Pediatric cardiology is currently a rapidly growing subspecialty. Children with more or less complex heart pathologies are now benefitting from increasingly complex and less invasive procedures thanks to the interventional cardiology branch. Simple conditions, such as atrial or ventricular communications, coarctation of the aorta, or patent ductus arteriosus, can be easily managed in a percutaneous manner. Furthermore, Blalock–Taussig shunts can now be closed in children developing pulmonary over-circulation-related heart failure symptoms, right ventricular outflow obstruction encountered in Fallot tetralogy can be interventionally managed, and inferior vena cava filters can be placed in selected patients. Implantable cardioverter-defibrillators have clearer use guidelines while cardiac resynchronization is usually reserved for dilated cardiomyopathies, as a bridge to transplantation.

## Figures and Tables

**Figure 1 diagnostics-12-01189-f001:**
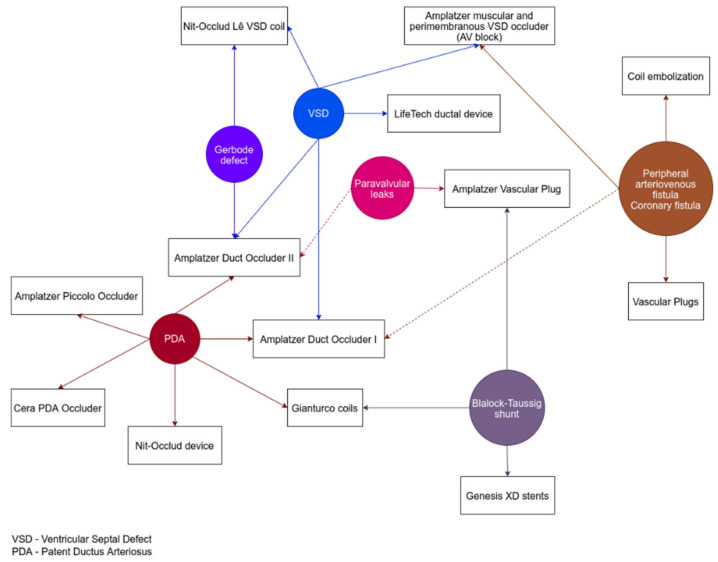
Devices used in systemic to pulmonary shunts, Gerbode defect, and paravalvular leaks.

**Figure 2 diagnostics-12-01189-f002:**
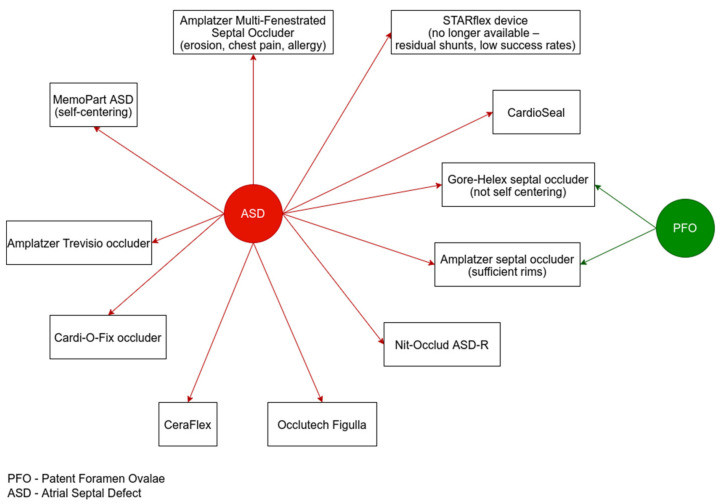
Routinely used devices in interatrial communications.

**Figure 3 diagnostics-12-01189-f003:**
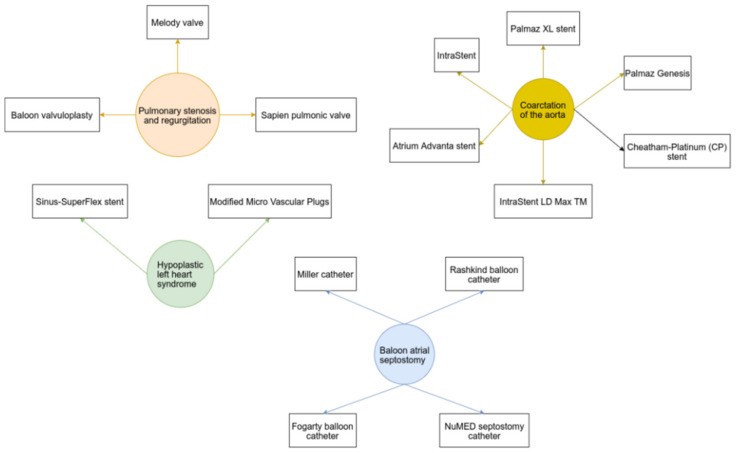
Devices used in various congenital heart defects.

**Table 1 diagnostics-12-01189-t001:** Frequency of the use of specific types of devices [17].

Frequently Used Devices	Infrequently Used Devices
Dilatation balloons	Rotational coronary atherectomy catheters
Occlusion devices	Inferior vena cava filters
Embolization coils	
Stents	
Septostomy catheters	
Septostomy balloons	
Guide wires	
Radiofrequency perforation wires	
Endovascular stents	
Cutting balloons	
Duct occluders	
Vascular plugs

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
