# Peer review of "An Up-to-Date Narrative Review on Congenital Heart Disease Percutaneous Treatment in Children Using Contemporary Devices"

_diagnostics, 2022, doi:10.3390/diagnostics12051189_

Round 1

Reviewer 1 Report

For the last paragraph in the introduction, it is recommended to have some summary statements on the existing stage of relevant development and their barriers.

In the last sentence of the introduction, it is recommended to include statements that layout the thematic framework and domain for the narrative review.

A concise informative content to justify that device was frequently/infrequency.

Figure 1 should be broken into 2 or more sub-figure in order to avoid arrows crossing each other.

Most of the device manufacturers' information or citation in section 5 novel devices are not available.

Before conclusion, the authors may indicate their perspective and recommendation for future work or development.

Author Response

We would like to thank the Reviewer and the Editorial board for all the positive remarks regarding our work. We are delighted to hear that the Editor / Reviewer observed the quality of the manuscript and the in-depth analysis of the subject.

We assure the EIC that we have read carefully every suggestion from this decision letter and tried our best to improve the quality of the document accordingly.

Q1: For the last paragraph in the introduction, it is recommended to have some summary statements on the existing stage of relevant development and their barriers.

Answer 1:

            Thank you for your excellent point!

            The Reviewer is right that it is required supplementary information regarding the barriers in pediatric interventional cardiology and the existing stage of relevant emerging devices. In consequence, we provided an additional paragraph in the introduction section, as the Reviewer suggested:

            “Some important barriers in pediatric transcatheter treatment innovations were de-scribed [15]. One of the most common barriers constitutes reluctance to testing and validation of novel devices in this pediatric population due to a greater concern re-garding legal or ethical aspects of the research. Also, conflicting knowledge related to reccomendations for development and manufacturing of pediatric devices could delay the evolution of percutaneous treatment [15]. Notwithstanding all the difficulties and barriers, novel pediatric devices emerged as a consequence of technological progress [16]”

Q2: In the last sentence of the introduction, it is recommended to include statements that layout the thematic framework and domain for the narrative review.

Answer 2:

            Thank you for the point raised!

            We agree with the Reviewer that additional statements concerning thematic framework and domain of the narrative review will enhance reading and understanding. Following the Reviewer recommendations, we modified the manuscript and provided additional statements in the introduction section:

            “We aimed to review the literature and to investigate the feasibility and efficacy of transcatheter interventions and contemporary implantable devices for congenital heart disease management in children. In consequence, we provided an updated narrative review integrating various percutaneous treatment techniques and devices used in pediatric interventional cardiology”

Q3: A concise informative content to justify that device was frequently/infrequency.

Answer 3:

            Thank you for your great point!

            The Reviewer is right regarding the need of an informative content to justify the frequency of the devices used. Therefore, we provided concise additional data from a clinical study on 473 pediatric patients:

            “In one study which enrolled 473 patients, the most used devices for transcatheter treatment were dilation balloons, occlusion devices, embolization coils and stents. Other percutaneous devices were used relatively uncommon [18]”.

Q4: Figure 1 should be broken into 2 or more sub-figure in order to avoid arrows crossing each other.

Answer 4:

Thank you very much for your suggestions.

We splitted the Figure as requested. We also inserted some colours.

Q5: Most of the device manufacturers' information or citation in section 5 novel devices are not available.

Answer 5:

            Great point indeed!

References added as requested!

Q6: Before conclusion, the authors may indicate their perspective and recommendation for future work or development.

Answer 6:

Thank you very much for your indications.

We added the following paragraph, quote:

  1. Perspectives.

This paper opens the pathway to future research related to pediatric cardiology. Future work should not only include a thorough follow-up on the novel devices discussed here but also on new interventional technique development and detailed research on nanotechnologies applicable to the newborn, infant, and pediatric cardiac malformations.”

Reviewer 2 Report

This review provides a current and up-to-date overview on the applicability of contemporary devices on percutaneous interventions in children with congenital heart disease. The authors did a great job in putting together the recent findings on different interventions and devices used for congenital heart disease management in children.

Specific comments:

  1. The authors discuss about percutaneous device treatment for congenital heart defect in children in the start of second paragraph (line 46), however there is no clear explanation to what percutaneous device treatment is. Please include 1-2 lines to define percutaneous device treatment.
  2. There is lack of linkage between the two paragraphs of “Amplatzer devices….” (Line 53-58) and “Balloon atrial septostomy….” (Line 59-64) with the paragraph before. Are the two devices linked to percutaneous device treatment? How are they different from each other?
  3. I found Figure 1 has too much information and it is not clear to me what is the core message here, and what is represented by ‘ASD’, ‘PFO’ etc.). Can the authors use color-coding to effectively group the different devices? Please also make sure the size of the texts in the figure are standardized (some texts are much smaller than others).
  4. For paragraph under the heading of “4. Specific interventions in pediatric cardiology”, can the authors make sure to have each subsection’s title consistent to naming the types of congenital heart disease follow by the intervention name? For example, subheading of 4.3 Aortic coarctation – percutaneous interventions, 4.5 Ventricular Septal Defect… etc (to avoid reader’s confusion)
  5. It’d be nice if the authors could include a summary figure under each sub-section of 4.1, 4,2 -4.13 of different intervention techniques.

Minor comments:

I do not think paragraph “2. Literature review methodology” (Line 68-80) is necessary to be included in the review.

Author Response

This review provides a current and up-to-date overview on the applicability of contemporary devices on percutaneous interventions in children with congenital heart disease. The authors did a great job in putting together the recent findings on different interventions and devices used for congenital heart disease management in children.

We would like to thank the Reviewer and the Editorial board for all the positive remarks regarding our work. We are delighted to hear that the Editor / Reviewer observed the quality of the manuscript and the in-depth analysis of the subject.

We assure the EIC that we have read carefully every suggestion from this decision letter and tried our best to improve the quality of the document accordingly.

Specific comments:

  1. The authors discuss about percutaneous device treatment for congenital heart defect in children in the start of second paragraph (line 46), however there is no clear explanation to what percutaneous device treatment is. Please include 1-2 lines to define percutaneous device treatment.

Answer 1:

Thank you for your excellent point!

The Reviewer is right that there is no clear explanation for specific context of percutaneous device treatment. Therefore, we provided an explanation in the second paragraph, as suggested by the Reviewer:

      “Percutaneous device treatment opened an entirely new era for congenital heart defect children, including those with isolated heart defect (pulmonary valve stenosis, patent ductus arteriosus – PDA, aortic coarctation, isolated inter-ventricular and inter-atrial communications), as well as those with complex heart disease (hypoplastic left heart syndrome – HLHS and tetralogy of Fallot – ToF). Percutaneous device treatment refers to transcatheter management of congenital heart defects using approved devices.”.

  1. There is lack of linkage between the two paragraphs of “Amplatzer devices….” (Line 53-58) and “Balloon atrial septostomy….” (Line 59-64) with the paragraph before. Are the two devices linked to percutaneous device treatment? How are they different from each other?

Answer 2:

Thank you for your excellent question!

The two paragraphs mentioned by the Reviewer are different from each other, despite the fact that they are linked to percutaneous device treatment. In the first mentioned paragraph we referred to percutaneous closure of atrial septal defect and patent ductus arteriosus using Amplatzer type devices.

However, in the second mentioned paragraph we described atrial septostomy, as a percutaneous method to create an inter-atrial communication in complex heart disease and pulmonary arterial hypertension, such as hypoplastic left heart syndrome, tricuspid or pulmonary valve atresia.

Therefore, although both paragraphs describe percutaneous treatment techniques, they are different as the first one addresses trascatheter closure of septal defects, while the other refers to creating an inter-atrial communication to preserve systemic blood flow.

  1. I found Figure 1 has too much information and it is not clear to me what is the core message here, and what is represented by ‘ASD’, ‘PFO’ etc.). Can the authors use color-coding to effectively group the different devices? Please also make sure the size of the texts in the figure are standardized (some texts are much smaller than others).

Answer 3:

      Thank you for your indications!

      We changed the Figure into more small Figures. We used colours as requested.

      We paid attention to the fonts and size.

  1. For paragraph under the heading of “4. Specific interventions in pediatric cardiology”, can the authors make sure to have each subsection’s title consistent to naming the types of congenital heart disease follow by the intervention name? For example, subheading of 4.3 Aortic coarctation – percutaneous interventions, 4.5 Ventricular Septal Defect… etc (to avoid reader’s confusion).

Answer 4:

      Than you for you great point!

      The Reviewer is right that different sub-headings titles contributes to reader’s confusion and might dilute the understanding.

Following the Reviewer recommendation, we modified individual subsection titles.

  1. It’d be nice if the authors could include a summary figure under each sub-section of 4.1, 4,2 -4.13 of different intervention techniques.

Answer 5:

      Thank you very much! We changed the Figure and splitted into different sub-sections as suggested.

Minor comments:

I do not think paragraph “2. Literature review methodology” (Line 68-80) is necessary to be included in the review.

Answer 6:

            Thank you for the point raised!

            We agree with the Reviewer that a literature review methodology is not mandatory in case of narrative reviews. Following the Reviewer suggestion, we removed the methodology section from the manuscript.